# Structural basis of ABCF-mediated resistance to pleuromutilin, lincosamide, and streptogramin A antibiotics in Gram-positive pathogens

Caillan Crowe-McAuliffe[1,7], Victoriia Murina [2,3,7], Kathryn Jane Turnbull[2,3], Marje Kasari[4], Merianne Mohamad[5], Christine Polte[1], Hiraku Takada [2,3], Karolis Vaitkevicius[2,3], Jörgen Johansson[2,3], Zoya Ignatova[1], Gemma C. Atkinson [2], Alex J. O'Neill[5], Vasili Hauryliuk [2,3,4,6✉] & Daniel N. Wilson [1✉]

Target protection proteins confer resistance to the host organism by directly binding to the antibiotic target. One class of such proteins are the antibiotic resistance (ARE) ATP-binding cassette (ABC) proteins of the F-subtype (ARE-ABCFs), which are widely distributed throughout Gram-positive bacteria and bind the ribosome to alleviate translational inhibition from antibiotics that target the large ribosomal subunit. Here, we present single-particle cryo-EM structures of ARE-ABCF-ribosome complexes from three Gram-positive pathogens: *Enterococcus faecalis* LsaA, *Staphylococcus haemolyticus* VgaA_LC and *Listeria monocytogenes* VgaL. Supported by extensive mutagenesis analysis, these structures enable a general model for antibiotic resistance mediated by these ARE-ABCFs to be proposed. In this model, ABCF binding to the antibiotic-stalled ribosome mediates antibiotic release via mechanistically diverse long-range conformational relays that converge on a few conserved ribosomal RNA nucleotides located at the peptidyltransferase center. These insights are important for the future development of antibiotics that overcome such target protection resistance mechanisms.

[1] Institute for Biochemistry and Molecular Biology, University of Hamburg, Hamburg, Germany. [2] Department of Molecular Biology, Umeå University, Umeå, Sweden. [3] Laboratory for Molecular Infection Medicine Sweden (MIMS), Umeå University, Umeå, Sweden. [4] University of Tartu, Institute of Technology, Tartu, Estonia. [5] Astbury Centre for Structural Molecular Biology, School of Molecular & Cellular Biology, Faculty of Biological Sciences, University of Leeds, Leeds, UK. [6] Department of Experimental Medical Science, Lund University, Lund, Sweden. [7] These authors contributed equally: Caillan Crowe-McAuliffe, Victoriia Murina. ✉email: vasili.hauryliuk@med.lu.se; Daniel.Wilson@chemie.uni-hamburg.de

The bacterial ribosome is a major antibiotic target[1]. Despite the large size of the ribosome, and the chemical diversity of ribosome-targeting small compounds, only a few sites on the ribosome are known to be bound by clinically used antibiotics. On the 50S large ribosomal subunit, two of the major antibiotic-binding sites are the peptidyl transferase centre (PTC) and the nascent peptide exit tunnel. The PTC is targeted by pleuromutilin, lincosamide and streptogramin A ($PLS_A$) antibiotics, as well as phenicols and oxazolidinones[2–6]. Representatives of macrolide and streptogramin B classes bind at adjacent sites at the beginning of the nascent peptide exit tunnel[3,5]. In contrast to the macrolides and streptogramin B antibiotics that predominantly inhibit translation during elongation[7], the $PLS_A$ antibiotics overlap with the amino acids attached to the CCA-ends of the A- and/or P-site of tRNAs and trap ribosomes during or directly after initiation[8–10]. This is highlighted by the recent usage of the pleuromutilin retapamulin to identify translation initiation sites in Ribo-Seq experiments[8].

Many mechanisms have evolved to overcome growth inhibition by such antibiotics in bacteria, among them target protection mediated by a subset of ABC family of proteins[11]. ATP-binding cassette (ABC) ATPases are a ubiquitous superfamily of proteins found in all domains of life, best-known as components of membrane transporters[12,13]. A typical ABC transporter contains two nucleotide-binding domains (NBDs), each of which contribute one of two faces to an ATP-binding pocket, as well as transmembrane domains[14]. Some sub-groups of ABC proteins, however, lack membrane-spanning regions and have alternative cytoplasmic functions, such as being involved in translation[15–17]. For example, in eukaryotes Rli1/ABCE1 is a ribosome splitting factor involved in recycling after translation termination, and the fungal eEF3 proteins bind the ribosome to facilitate late steps of translocation and E-site tRNA release[18,19]. The F-type subfamily of ABC proteins, which are present in bacteria and eukaryotes, contain at least two NBDs separated by an α-helical interdomain linker and notably lack transmembrane regions[20–22].

One group of bacterial ABCFs, which are termed antibiotic resistance (ARE) ABCFs[23], confer resistance to antibiotics that bind to the 50S subunit of the bacterial ribosome[11,21,24,25]. Characterized ARE-ABCFs are found predominantly in Gram-positive bacteria, including human and animal pathogens, typically have a restricted host specificity, and can be further divided into eight subfamilies[11,20,26]. Although initially thought to act as part of efflux systems[27,28], these proteins were subsequently shown instead to bind the ribosome, oppose antibiotic binding, and to reverse antibiotic-mediated inhibition of translation in vitro[29].

Phylogenetic analyses indicate that ARE-ABCFs may have arisen multiple times through convergent evolution, and that antibiotic specificity can be divergent within a related subgroup[20]. Classified by the spectrum of conferred antibiotic resistance, ARE-ABCFs can be categorized into eight subfamilies with three different resistance spectra[20,25]:

1. A highly polyphyletic group of ARE-ABCFs that confer resistance to the PTC-binding $PLS_A$ antibiotics (ARE1, ARE2, ARE3, ARE5 and ARE6 subfamilies). The most well-studied representatives are VmlR, VgaA, SalA, LmrC and LsaA[26,30–33]. Additionally, a lincomycin-resistance ABCF that belongs to this group, termed Lmo0919, has been reported in *Listeria monocytogenes*[34–36].

2. ARE-ABCFs that confer resistance to antibiotics that bind within the nascent peptide exit channel (a subset of the ARE1 subfamily, and ARE4). The most well-studied representatives are Macrolide and streptogramin B resistance (Msr) proteins[28,37,38].

3. Poorly experimentally characterized ARE-ABCF proteins belonging to subfamilies ARE7 (such as OptrA) and ARE8 (PoxtA). These resistance factors confer resistance to phenicols and oxazolidinones that bind in the PTC overlapping with the $PLS_A$-binding site[11,39,40] and are spreading rapidly throughout bacteria in humans and livestock by horizontal gene transfer[41–44].

Additionally, several largely unexplored groups of predicted novel ARE-ABCFs are found in high-GC Gram-positive bacteria associated with antibiotic production[20].

So far, two structures of ARE-ABCFs bound to the 70S ribosome have been determined[24,38,45]. In each instance, the ARE-ABCF interdomain linker extends from the E-site-bound NBDs into the relevant antibiotic-binding site in the ribosome, distorting the P-site tRNA into a non-canonical state located between the P and E sites. The tip of the interdomain linker—termed the antibiotic resistance determinant (ARD) in ARE-ABCFs—is not well conserved among (and sometimes not even within) subfamilies, and mutations in this region can abolish activity as well as change antibiotic specificity. Mutagenesis indicates that both steric overlap between the ARD and the antibiotic, as well as indirect reconfiguration of the rRNA and the antibiotic-binding site, may contribute to antibiotic resistance[24,38,45,46]. Non-ARE ribosome-associated ABCFs that do not confer resistance to antibiotics—such as EttA—tend to have relatively short interdomain linkers that contact and stabilize the P-site tRNA[22,47]. ARE-ABCFs that confer resistance to $PLS_A$ antibiotics (such as VmlR) have extensions in the interdomain linker that allow them to reach into the antibiotic-binding site in the PTC[45]. The longest interdomain linkers belong to ARE-ABCFs that confer resistance to macrolides and streptogramin B antibiotics (e.g. MsrE), and such linkers can extend past the PTC into the nascent peptide exit tunnel[38]. The length of the bacterial ABCF ARD generally correlates with the spectrum of conferred antibiotic resistance. Notable exceptions to this pattern are OptrA and PoxtA ARE-ABCF which have short interdomain linkers, yet still confer resistance to some PTC-binding antibiotics[39,40], while typically PTC-protecting ARE-ABCFs such as VmlR, LsaA and VgaA typically have comparatively long interdomain linkers[46,48].

The available ARE-ABCF-ribosome structures were generated by in vitro reconstitution. *Pseudomonas aeruginosa* MsrE, which confers resistance to tunnel-binding macrolides and streptogramin B antibiotics (that inhibit translation elongation) was analysed bound to a heterologous *Thermus thermophilus* initiation complex[38]. *Bacillus subtilis* VmlR, which confers resistance to $PLS_A$ antibiotics that bind in the PTC (which stall translation at initiation), was analysed in complex with a *B. subtilis* 70S ribosome arrested during elongation by the presence of a macrolide antibiotic[33,45]. Structures of native physiological complexes (such as those generated using pull-down approaches from the native host) are currently lacking.

In this work, we systematically characterize the antibiotic resistance specificity and determine the structure of three in vivo formed ARE-ABCF-70S ribosome complexes using affinity chromatography and cryo-electron microscopy (cryo-EM). Our study focusses on ARE-ABCFs that confer resistance to $PLS_A$ antibiotics in clinically relevant Gram-positive pathogens, namely, the ARE3 representative *Enterococcus faecalis* LsaA[30], as well as the ARE1 representatives *Listeria monocytogenes* Lmo0919 (refs. [34–36])—which we have termed VgaL—and the well-characterized $VgaA_{LC}$ protein, initially isolated from *Staphylococcus haemolyticus*[26,36,46,49,50]. *Staphylococcus* and *Enterococcus* are commensal organisms that are prevalent in diverse healthcare-associated infections, and antibiotic resistance is

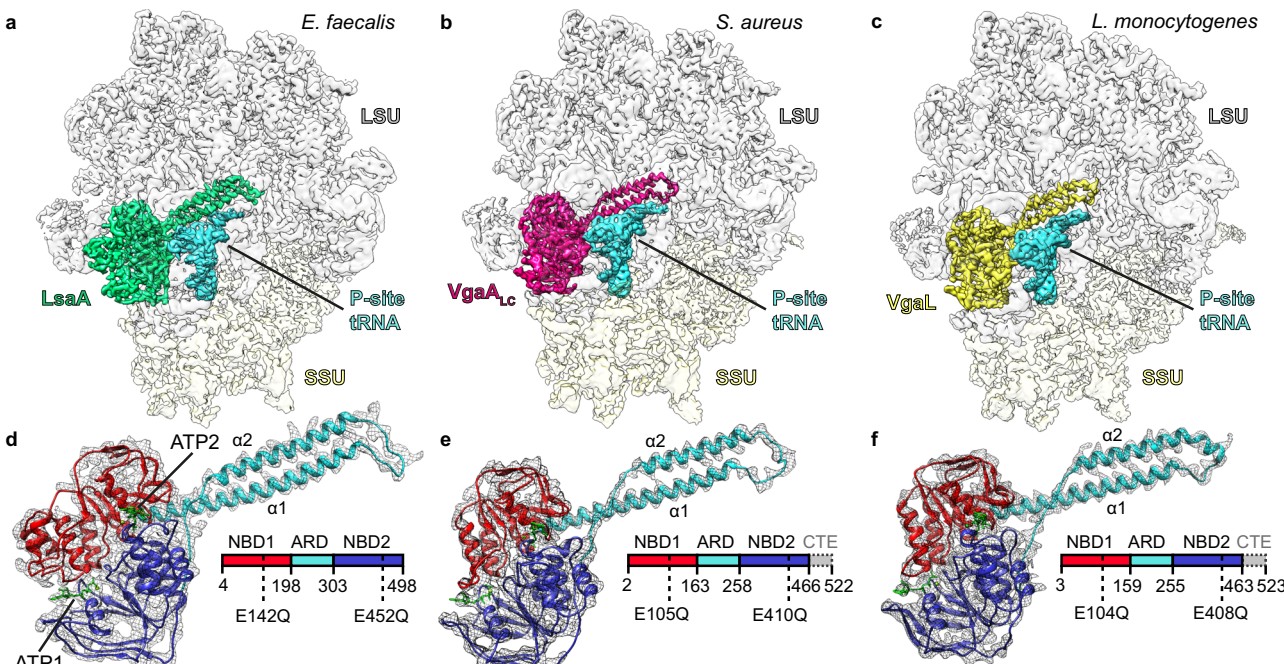

**Fig. 1 Cryo-EM structures of ARE-ABCF–ribosome complexes. a–c** Cryo-EM maps with isolated densities for **a** *E. faecalis* LsaA (green), **b** *S. aureus* VgaA_LC (magenta), **c** *L. monocytogenes* VgaL (yellow) as well as P-site tRNA (cyan), small subunit (SSU, yellow) and large subunit (LSU, grey). **d–f** Density (grey mesh) with molecular model for **d** LsaA, **e** VgaA_LC and **f** VgaL, coloured according to domain as represented in the associated schematics: nucleotide-binding domain 1 (NBD1, red), antibiotic-resistance domain (ARD, cyan), nucleotide-binding domain 2 (NBD2, blue) and C-terminal extension (CTE, grey, not modelled). α1 and α2 indicate the two α-helices of the ARD interdomain linker. In **d–f**, the ATP nucleotides are coloured green.

spreading through these species[51–54]. *L. monocytogenes* is a foodborne pathogen that poses a particular risk to pregnant women and immunocompromised patients[55]. Our structures, supported by extensive mutagenesis experiments, provide insight into the mechanism by which these distinct ARE-ABCFs displace antibiotics from their binding site on the ribosome to confer antibiotic resistance.

## Results

**Cryo-EM structures of in vivo formed ARE-ABCF-70S complexes**. To obtain in vivo formed ARE-ABCF-70S complexes, we expressed C-terminally FLAG$_3$-tagged ATPase-deficient EQ$_2$ variants of *E. faecalis* LsaA, *S. aureus* VgaA_LC and *L. monocytogenes* VgaL in their corresponding native host bacterial species. The FLAG$_3$ tag was used for affinity purification of each protein locked onto their respective ribosomal target. The ARE-ABCFs co-migrated with the 70S fraction through sucrose gradients—with the complex further stabilized in the presence of ATP in the case of LsaA and VgaA_LC—and co-eluted with ribosomal proteins after affinity purification (Supplementary Figs. 1–3).

The resulting in vivo formed complexes were characterized by single-particle cryo-EM (see 'Methods'), yielding ARE-70S complexes with average resolutions of 2.9 Å for *E. faecalis* LsaA, 3.1 Å for *S. aureus* VgaA_LC and 2.9 Å for *L. monocytogenes* VgaL (Fig. 1a–c, Supplementary Table 4 and Supplementary Figs. 4–6). In each instance, the globular NBDs of the ARE-ABCF were bound in the E-site, and the α-helical interdomain linker, consisting of two α-helices (α1 and α2) and the ARD loop, extended towards the PTC (Fig. 1a–c). Additionally, a distorted tRNA occupied the P-site (Fig. 1a–c), similarly to what was observed previously for *P. aeruginosa* MsrE and *B. subtilis* VmlR[38,45]. For the LsaA and VgaL samples, occupancy of the factor on the ribosome was high, with >95% and ~70% of picked ribosomal particles containing LsaA and VgaL, respectively

(Supplementary Figs. 4 and 6). By contrast, VgaA_LC had lower occupancy (~60%), implying that the factor dissociated after purification and/or during grid preparation (Supplementary Fig. 5). In silico 3D classification revealed that the major class not containing VgaA_LC in the dataset was a 70S ribosome with P-tRNA, which could also be refined to an average resolution of 3.1 Å (Supplementary Fig. 5). Generally, the 50S ribosomal subunit and ARE-ABCF interdomain linkers were well-resolved (Fig. 1d–f and Supplementary Figs. 4–6). While ARE-ABCF NBDs, occupying the E-site, had a lower resolution—especially in the regions that contact the ribosomal L1 stalk and the 30S subunit—the density was nonetheless sufficient to dock and adjust homology models in each instance (Fig. 1d–f and Supplementary Figs. 4–6). Densities corresponding to the 30S subunits were of lower quality, indicating flexibility in this region, but, with multibody refinement, were nonetheless sufficient to build near-complete models of each ribosome. Density between NBD1 and NBD2 of each ARE was most consistent with the presence of two molecules of ATP (or another NTP) and a cation, which we tentatively assigned as ATP-1, ATP-2 and magnesium, respectively. Nonetheless, the density in this region was not sufficiently detailed to model this region de novo and caution is warranted in interpreting exact geometries from the model (Fig. 1d–f and Supplementary Fig. 7). Interestingly, the density for the nucleobase of ATP-1 bound in the peripheral nucleotide-binding site of each ARE-ABCF was particularly poor (Supplementary Fig. 7), consistent with the relaxed nucleotide specificity of these proteins, i.e., the ability of ARE-ABCFs to hydrolyze other nucleotides, such as CTP, UTP and GTP[56].

By comparison to structures of other ABC proteins, the NBDs adopted a closed conformation bound tightly to each nucleotide (Supplementary Fig. 8). In each ARE-bound 70S structure, the ribosomal small subunit was in a semi-rotated state, although this varied between AREs, with the LsaA- and VgaL-bound ribosomes more rotated than VgaA_LC-bound 70S (Supplementary Fig. 9a–d).

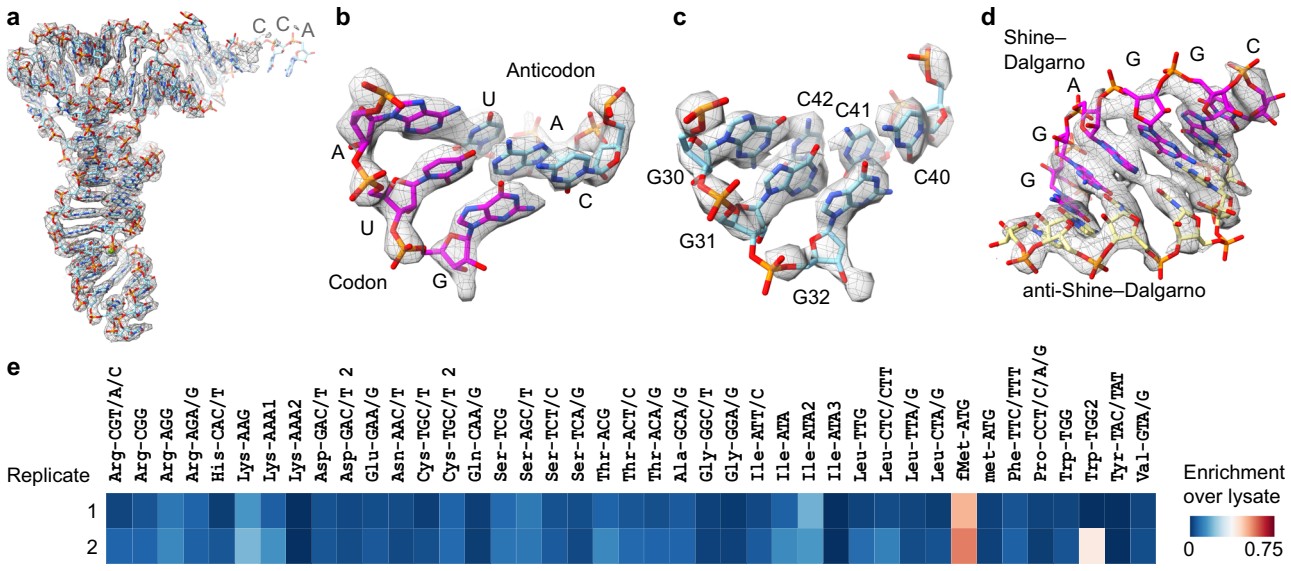

**Fig. 2 The LsaA-70S complex contains an initiator tRNA and SD-helix. a–d** Isolated density (grey mesh) with molecular models (sticks) for **a** initiator tRNA$^{fMet}$ (cyan), **b** interaction between AUG start codon of the mRNA (magenta) and anticodon of initiator tRNA$^{fMet}$ (cyan) in the P-site, **c** three G:C base pairs specific to the initiator tRNA$^{fMet}$ (cyan) and **d** helix formed between Shine-Dalgarno (SD) sequence of the mRNA (magenta) and anti-SD of the 16S rRNA (yellow). **e** Replicate tRNA microarray analysis of the LsaA-70S complex, illustrating the enrichment of initiator tRNA$^{fMet}$ in the LsaA-70S complex over the lysate. Confidence intervals between replicates were 92%.

In each ARE-ABCF-70S map, the P-site tRNA was distorted compared to a classic P-site tRNA, resulting in a substantial shift of the acceptor stem away from the PTC (Supplementary Fig. 9e–h), as observed previously for MsrE and VmlR[38,45]. In each case, the distorted P-site tRNA was rotated compared to a classic P-site tRNA (21–29°), possibly due to a defined interaction of the tRNA elbow with the NBD2 of the ARE (Supplementary Fig. 9i–k). The CCA 3′ end was particularly disordered, precluding any additional density corresponding to an amino acid or nascent chain from being modelled, although the approximate path could be traced in low-pass-filtered maps (Fig. 1a–c and Supplementary Figs. 4–6). We have used our high-resolution maps to present a model of the ribosome from the Gram-positive pathogen *L. monocytogenes* and update the model of the *S. aureus* ribosome[57]. Our models of the *E. faecalis* and *S. aureus* ribosomes are in good agreement with those recently described[58,59].

**LsaA, VgaA$_{LC}$ and VgaL bind to translation initiation states.** In each cryo-EM map, the P-site tRNA body was sufficiently well-resolved to unambiguously assign the density to initiator tRNA$^{fMet}$ on the basis of (i) general fit between sequence and density, (ii) the well-resolved codon–anticodon interaction and (iii) a characteristic stretch of G:C base pairs found in the anticodon stem loop of tRNA$^{fMet}$ (Fig. 2a–c). Additionally, in the small subunit mRNA exit tunnel, density corresponding to a putative Shine-Dalgarno–anti-Shine-Dalgarno helix was observed, consistent with the ARE-ABCF binding to an initiation complex containing tRNA$^{fMet}$ (Fig. 2d). LsaA–*E. faecalis* 70S samples were further analysed with a custom tRNA microarray, which confirmed tRNA$^{fMet}$ was the dominant species found in the sample (Fig. 2e). Collectively, these observations indicate that in our structures the majority of the ARE-ABCFs are bound to 70S translation initiation complexes. While the initiation state is also the state that would result from PLS$_A$ inhibition, we note that our complexes were formed in the absence of an antibiotic. Thus, in our experimental set-up it is likely that the use of the EQ$_2$ mutants traps the ARE-ABCFs on initiation complexes due to the availability of the E-site.

Further examination of the LsaA-70S volume revealed weak density in the ribosomal A-site (Supplementary Fig. 4f), suggesting that some complexes had entered the elongation cycle. This was unexpected, as the distorted P-site tRNA is predicted to overlap with an accommodated A-site tRNA, although as noted would be compatible with a pre-accommodated A/T-tRNA[45]. A mask around the A-site was used for partial signal subtraction, and focused 3D classification was used to further sub-sort the LsaA-70S volume. One class, containing approximately one-third of the particles, was shown to indeed contain a tRNA in the A-site (Supplementary Figs. 4 and 10a). This tRNA was poorly resolved, suggesting flexibility, and was slightly rotated compared to a canonical, fully accommodated A-site tRNA, and, as for the P-site tRNA, the acceptor stem was significantly disordered and displaced (Supplementary Fig. 10b, c). This state likely reflects an incomplete or late-intermediate accommodation event, as observed previously when translation is inhibited by PTC-binding antibiotics hygromycin A or A201A, both of which were shown to sterically exclude the acceptor stem of a canonical A-site tRNA[60]. A very weak density corresponding to an A-site tRNA was also observed in VgaA$_{LC}$ and VgaL volumes, but sub-classification was unsuccessful for these datasets.

VgaA$_{LC}$ and VgaL, both of which belong to the ARE1 subfamily—although not LsaA, which belongs to the ARE3 subfamily—contain a short C-terminal extension (CTE) predicted to form two α-helices[20,45]. Although not conserved among all AREs, deletion of the CTE abolished antibiotic resistance in VmlR and reduced antibiotic resistance in VgaA, implying that this extension is necessary for function in some ARE-ABCFs[45,49]. Density for this region, which emanates from NBD2 and was located between ribosomal proteins uS7 and uS11, was present in the VgaA$_{LC}$-70S and VgaL-70S maps and was essentially consistent with the position of the VmlR CTE, although was not sufficiently resolved to create a model for this region. Although bound close to the mRNA exit channel, the CTEs of VgaA$_{LC}$ and VgaL did not contact the Shine-Dalgarno–anti-Shine-Dalgarno helix of the initiation complexes, indicating they are not critical for substrate recognition in these ARE-ABCFs (Supplementary Fig. 10d–f).

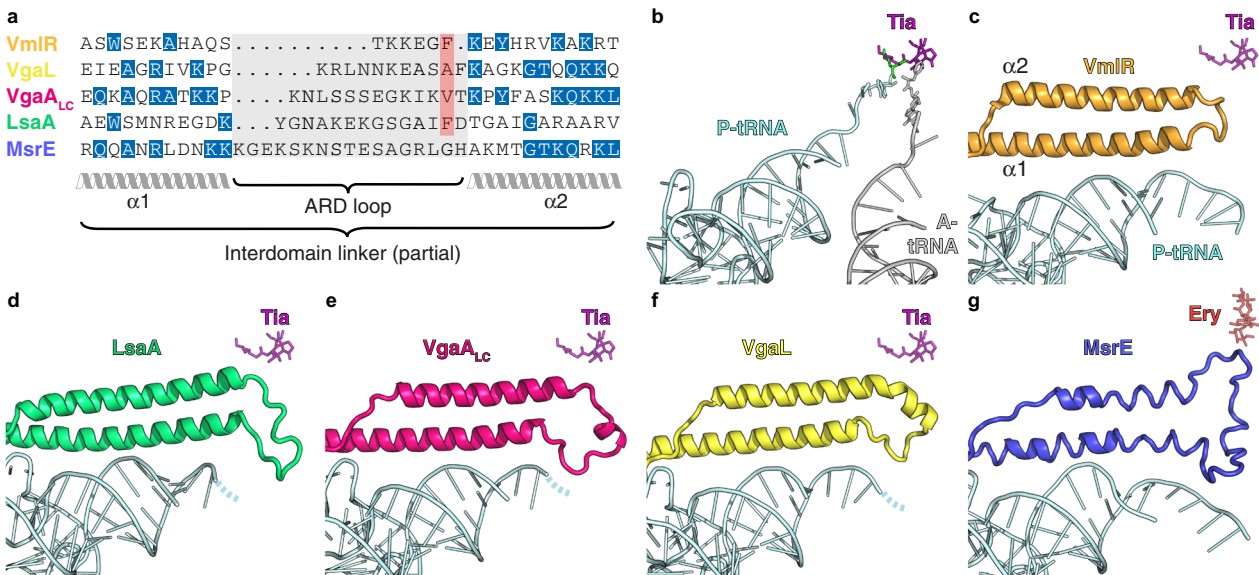

**Fig. 3 Comparison of the ARD loops of different ARE-ABCFs. a** The sequence length of the ARD loops differs significantly for VmlR, VgaL, VgaA$_{LC}$, LsaA and MsrE. Although the lack of sequence homology precludes accurate sequence alignment of the ARD loops, the red highlighted residues can be aligned structurally. Sequences were aligned with Clustal Omega and edited by hand to match the structures. **b–g** Comparison of the positions of **b** A-site tRNA (grey) and P-site tRNA (cyan) from pre-attack state (PDB 1VY4)[103], with shifted P-site tRNA (cyan) and ABCF ARD from ribosome complexes containing **c** VmlR (orange, PDB 6HA8)[45], **d** LsaA (green), **e** VgaA$_{LC}$ (magenta), **f** VgaL (yellow) and **g** MsrE (blue, PDB 5ZLU)[38]. In **b–g**, the relative position of either tiamulin (Tia, magenta, PDB 1XBP)[2] or erythromycin (Ery, red, PDB 4V7U)[5] has been superimposed. Dashed lines in **d–f** represent the likely path of the CCA end of the tRNA.

## The location and conformation of short and long ARDs on the ribosome.

The ARD loop, positioned between the two long α-helices that link the NBDs, is a critical determinant of antibiotic resistance[29,38,45,46,56]. Despite sharing a similar antibiotic specificity profile, the ARDs of LsaA, VgaA$_{LC}$, VgaL and VmlR are divergent in both amino acid composition and length, which is consistent with the polyphyletic nature of this group but precludes confident sequence alignment of this region (Fig. 3a). Despite such sequence divergence, the position of the ARDs on the ribosome is broadly similar in each instance (Fig. 3b–g). By comparison to tiamulin, which overlaps with the aminoacyl moieties of A- and P-tRNAs in the PTC[2,60], VmlR, LsaA, VgaA$_{LC}$ and VgaL are all positioned similarly on the ribosome, with the ARD backbone adjacent to the antibiotic-binding site (Fig. 3b–f). Compared to VmlR[45], the additional residues in the ARDs of LsaA, VgaA$_{LC}$ and VgaL extend away from the antibiotic-binding site, towards the CCA 3′ end of the distorted P-tRNA (Fig. 3c–f). By contrast, MsrE, which confers resistance to tunnel-binding antibiotics deeper in the ribosome, has a longer ARD that extends both past the PTC to approach the macrolide/streptogramin A-binding site, as well as towards the distorted P-tRNA (Fig. 3a, g)[38]. Thus, the length of the ARD does not necessarily provide insights into the extent to which the ARD will penetrate into the ribosomal tunnel and thus one cannot easily predict whether long ARDs will confer resistance to macrolide antibiotics.

## Position of the ARDs with respect to PLS$_A$ antibiotic-binding site.

We next made a careful comparison of the LsaA, VgaA$_{LC}$ and VgaL ARDs with the binding sites of relevant antibiotics within the PTC (Fig. 4a, b)[2,5,6,61]. For LsaA, the side chain of Phe257 overlapped with the binding sites of tiamulin, virginiamycin M and lincomycin, but was not close to erythromycin (Fig. 4a–c), consistent with the spectrum of antibiotic resistance conferred by this protein (Supplementary Table 1). In the VgaA$_{LC}$ ARD, Val219 was situated close to tiamulin and virginiamycin M, and had a modest predicted overlap with lincomycin (Fig. 4d).

Notably, in the closely related variant VgaA, which has a similar specificity with modestly higher resistance to tiamulin and virginiamycin M, residue 219 is a glycine, which we predict would not overlap with the PLS$_A$-binding site[46]. Thus, VgaA$_{LC}$ confers resistance to virginiamycin M and tiamulin despite the lack of overlap between the ARE-ABCF and the antibiotic-binding site (Supplementary Table 2). For VgaL, the closest residue to the PLS$_A$-binding site was Ala216, which had no predicted overlap with tiamulin, virginiamycin M or lincomycin (Fig. 4e). VgaL therefore confers resistance to lincomycin, virginiamycin M and tiamulin without directly overlapping the binding sites of these antibiotics. In summary, there was no general pattern of overlap or non-overlap with the PLS$_A$ binding sites among LsaA, VgaA$_{LC}$ and VgaL, and our structural evidence is not consistent with a steric overlap model of antibiotic egress.

## Mutational analysis of LsaA and VgaA$_{LC}$ ARDs.

Our models of the ARD loops allowed us to design and test mutants for capacity to confer antibiotic resistance. Because genetic manipulation in *Enterococcus faecalis* is difficult, and LsaA complements the *B. subtilis* Δ*vmlR* strain (Supplementary Fig. 11), we performed the mutational analysis of LsaA in the *B. subtilis* Δ*vmlR* background. When LsaA Phe257, which directly overlaps the PLS$_A$-binding site (Fig. 4c), was mutated to alanine, no change in resistance was observed (Supplementary Fig. 11). By contrast, mutation of Lys244, which is not situated close to the PLS$_A$-binding sites but forms a hydrogen bond with 23S rRNA G2251 and G2252 (*Escherichia coli* numbering is used for 23S rRNA nucleotides) of the P-loop (Supplementary Fig. 11), nearly abolished antibiotic resistance activity (Supplementary Fig. 12). Taken together, these observations indicate that LsaA does not confer resistance via simple steric occlusion, and that interactions with the P-loop may be required for positioning the LsaA ARD.

For VgaA$_{LC}$, extensive alanine mutations within the ARD were explored (Supplementary Table 2). As expected from the above analyses and natural variants, mutating Val219—the only residue in VgaA$_{LC}$ that sterically overlaps the PLS$_A$-binding site—did not

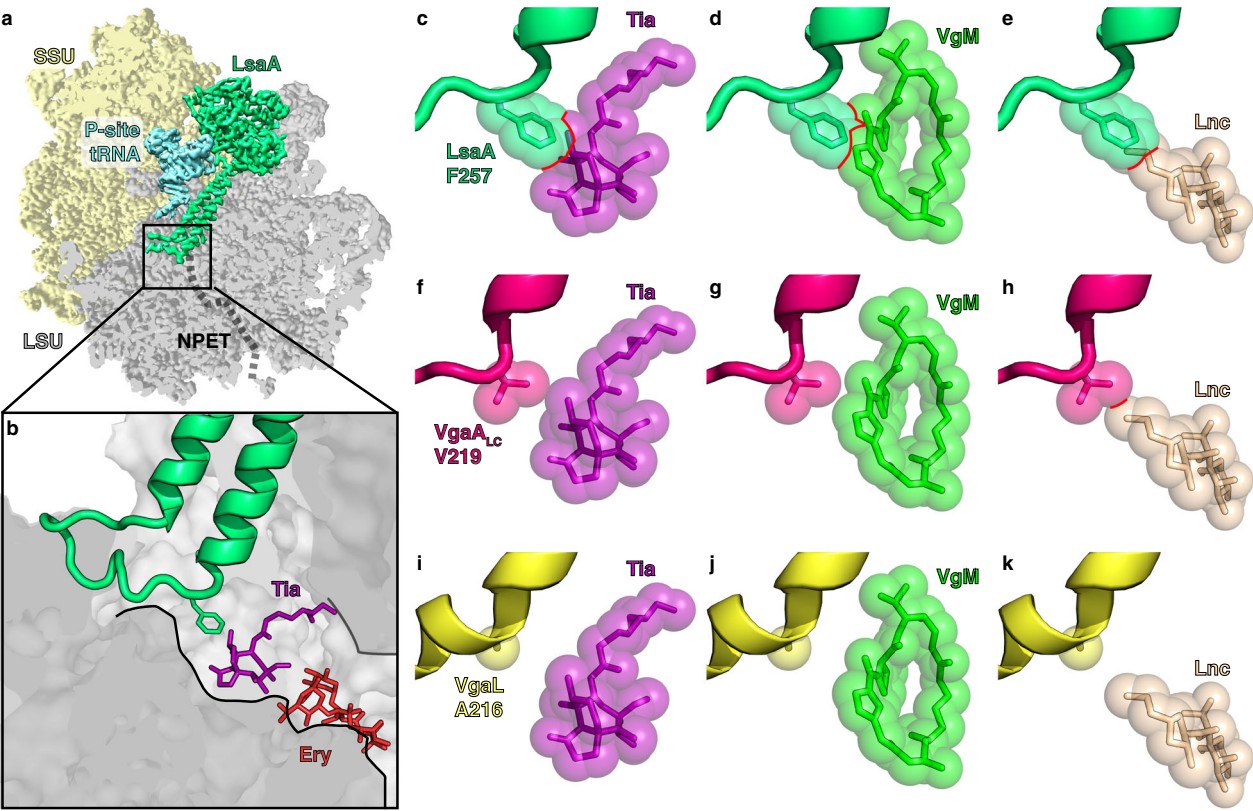

**Fig. 4 Interaction of LsaA, VgaA_LC and VgaL at the peptidyl transferase centre. a–b** LsaA and distorted P-site tRNA superimposed on a transverse section of the large subunit (LSU, grey) to reveal **a** the ARD of LsaA extending into the nascent polypeptide exit tunnel (NPET) and **b** the relative position of Phe257 of LsaA to tiamulin (Tia, purple, PDB 1XBP)[2] and erythromycin (Ery, red, PDB 4V7U)[5]. **c–k** Relative position of LsaA (green, top row, **c–e**), VgaA_LC (pink, middle row, **f–h**) and VgaL (yellow, bottom row, **i–k**) to tiamulin (Tia, purple, PDB 1XBP), virginiamycin M (VgM, lime, PDB 4U25)[61] and lincomycin (Lnc, tan, PDB 5HKV)[6]. When present, clashes in **c–k** are shown with red outlines.

affect the antibiotic resistance profile. Three residues at the beginning of α2, directly after the ARD loop, were required for resistance: Tyr223, which stacks with U2585 (part of the pleuromutilin-binding site); Phe224, which stacks with A2602 held in the centre of the ARD; and Lys227, which forms a hydrogen bond with the 5′ phosphate of C2601 (Supplementary Table 2). These residues do not overlap with the PLS_A-binding site, but may be required to position the ARD in the PTC to impede antibiotic binding, or for the folding of the ARD itself (Supplementary Fig. S12d–f). In the naturally variable VgaA_LC ARD loop, mutation of Ser213, which sits adjacent to U2506 and C2507 (Supplementary Fig. S12e), to alanine similarly reduced antibiotic resistance (Supplementary Table 2). Of note, mutating the most conserved residue among VgaA variants in this region, Lys218, did not substantially affect resistance (Supplementary Table 2)[62]. Extensive alanine substitutions in the surrounding residues that contact the 23S rRNA (Supplementary Fig. 12d–f) either did not affect, or had only a mild influence on, the antibiotic resistance conferred by this protein (Supplementary Table 2). In summary, mutation of VgaA_LC residues that interact with 23S rRNA nucleotides that form part of the PLS_A-binding pocket affected antibiotic-resistance activity.

**Modulation of the ribosomal antibiotic-binding site by ARE-ABCFs.** We next sought to explore how the ARDs of LsaA, VgaA_LC and VgaL affect the conformation of the ribosomal PTC. The 23S rRNA A2602, which is flexible in the absence of tRNAs and positioned between the P- and A-tRNAs during peptidyl transfer, is bound and stabilized by all structurally characterized ARE-ABCFs. In LsaA and VmlR, a tryptophan stacks and

stabilizes A2602 in a flipped position (Supplementary Fig. 13)[45], reminiscent of the stacking interaction between the equivalent rRNA nucleotide and Tyr346 of the yeast ABCF protein Arb1 observed in a structure of a ribosome-associated quality control complex[63]. In VgaA_LC, VgaL, and MsrE, A2602 is instead positioned within the ARD loop, interacting with multiple residues from the ARE (Supplementary Fig. 13)[38].

We have designated four regions within domain V of the 23S rRNA (Fig. 5a) as PTC loops 1–4 (PL1–4) that comprise the binding site for the A- and P-site tRNA (Fig. 5b), are close to the ARD of the ARE-ABCFs (Fig. 5c) and form the binding pocket for the PLS_A antibiotics (Fig. 5d–f). There is a significant overlap between nucleotides that form the PLS_A-binding pockets and nucleotides that are shifted when LsaA, VgaA_LC or VgaL are bound to the ribosome (Fig. 5a). While the ARE-ABCFs come close to PL1, they do not interact directly and the conformation of nucleotides within PL1 do not appear to be altered when comparing the ARE-ABCF and PLS_A conformations (Fig. 5g–i and Supplementary Fig. 14). An exception was a slight rotation of the A2062 nucleobase (Supplementary Fig. 14), which is most likely a consequence of drug binding rather than ARE engagement. By contrast, multiple rearrangements were evident in PL2 that appear to arise due to direct contact between the ARD loop of the ARE-ABCF and the backbone of 23S rRNA nucleotides A2451–A2452 within PL2 (Fig. 6a–d and Supplementary Fig. 15). Displacement of the backbone was largest (3.3–4.4 Å) upon LsaA binding, intermediate (3.1 Å) for VgaA_LC, and smallest (1.0 Å) for VgaL, and resulted in corresponding shifts in the position of the nucleobases that comprise the PLS_A-binding pocket (Fig. 6a–d and Supplementary Fig. 15).

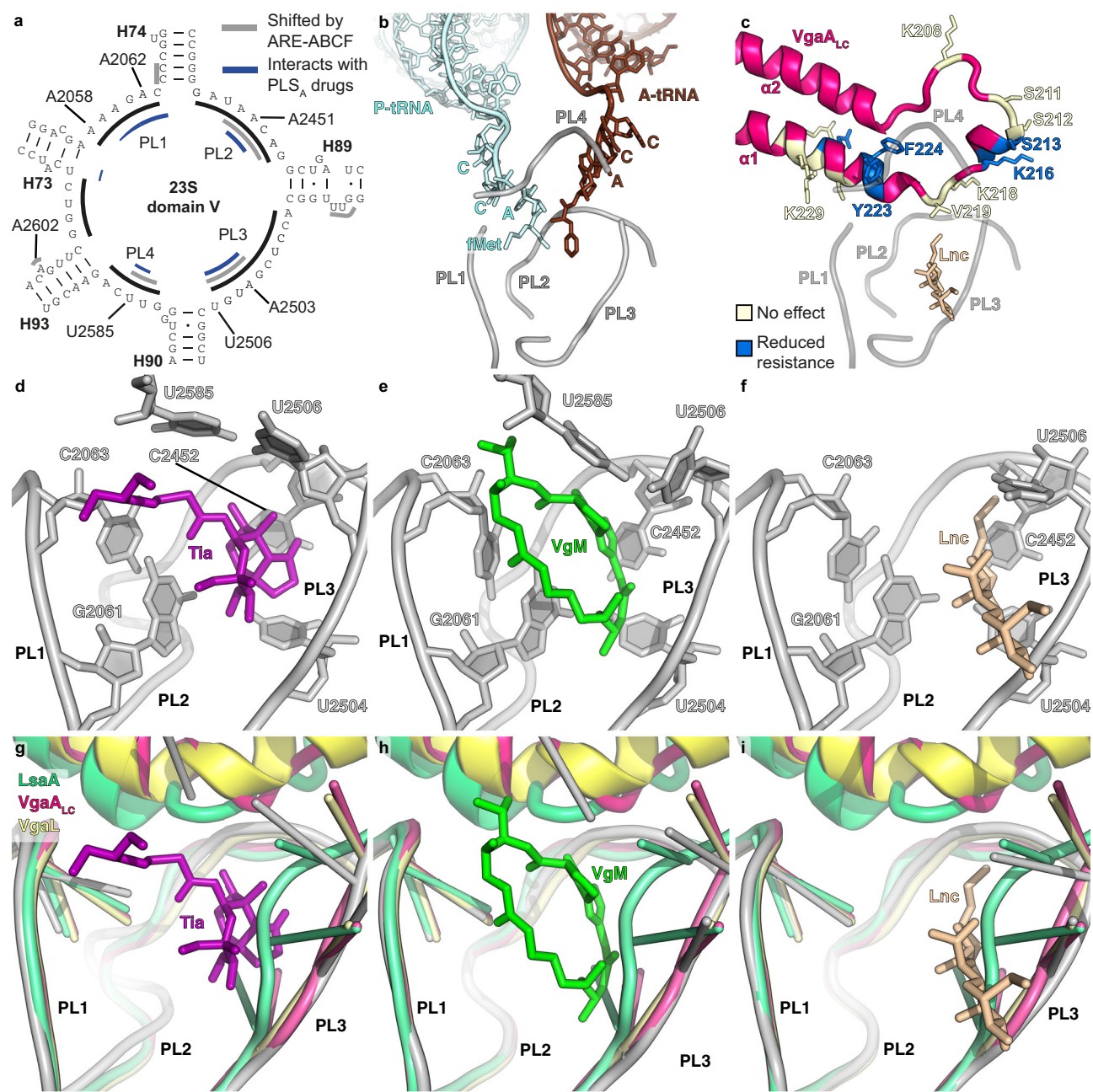

**Fig. 5 ARE-ABCF binding induces conformational changes at the PTC. a** Secondary structure of peptidyl transferase ring within domain V of the 23S rRNA, highlighting residues within PTC loops 1–4 (PL1–4) that comprise the binding site of PLS$_A$ antibiotics (blue) and/or undergo conformational changes upon ARE-ABCF binding (grey). **b** View of the PTC in the pre-peptidyl transfer state (PDB 1VY4)[103] with tRNAs and PLs 1–4 from **a** labelled. **c** Same view as **b**, except with the VgaA$_{LC}$ structure shown. For reference, lincomycin is also included (PDB 5HKV)[6]. Residues coloured yellow had no effect on resistance when mutated to alanine. For residues coloured blue, antibiotic resistance was significantly affected when mutated to alanine. **d**–**f** Binding site of **d** tiamulin (Tia, magenta, PDB 1XBP, 3.5 Å)[2], **e** virginiamycin M (VgM, green, PDB 4U25, 2.9 Å)[61] and **f** lincomycin (Lnc, tan, PDB 5HKV, 3.7 Å)[6] on the ribosome. **g**–**i** Comparison of conformations of rRNA nucleotides comprising the **g** Tia, **h** VgM and **i** Lnc binding site (shown as grey cartoon ladder representation), with rRNA conformations when LsaA (green), VgaA$_{LC}$ (magenta) or VgaL (yellow) are bound.

Unexpectedly, large changes were also observed in PL3, around nucleotides U2504–U2506, in the ARE-bound structures, despite the lack of contact between this region and the ARDs (Fig. 6e–h and Supplementary Fig. 16). Such shifts are likely a consequence of disturbances in PL2 since nucleotides within PL2 are in direct contact with nucleotides in PL3 (Fig. 6i). Specifically, the 23S rRNA nucleotides G2505 and U2506 in PL2 were shifted by 2.8-3.0 Å when comparing each ARE-bound 70S to the drug-bound states (Fig. 6e–h and Supplementary Fig. 17). Additionally, in the LsaA-bound 70S, U2504 was shifted such that it directly overlaps with the PLS$_A$-

binding site (Fig. 6f). The rearrangement of U2504 appears to arise because of a cascade of changes in PL2 due to LsaA binding, namely, A2453 of PL2 is shifted slightly away from the PTC and pairs with G2499 (instead of U2500), allowing C2452 (which normally pairs with U2504 and forms part of the PLS$_A$-binding pocket) to instead hydrogen bond with U2500. The relocation of C2452 frees U2504, and PL3 more generally, allowing it to reposition into the PLS$_A$-binding pocket upon LsaA binding (Fig. 6i, j).

U2585, which is part of PL4, forms part of the tiamulin (Fig. 6k) and virginiamycin M-binding site, but not that of

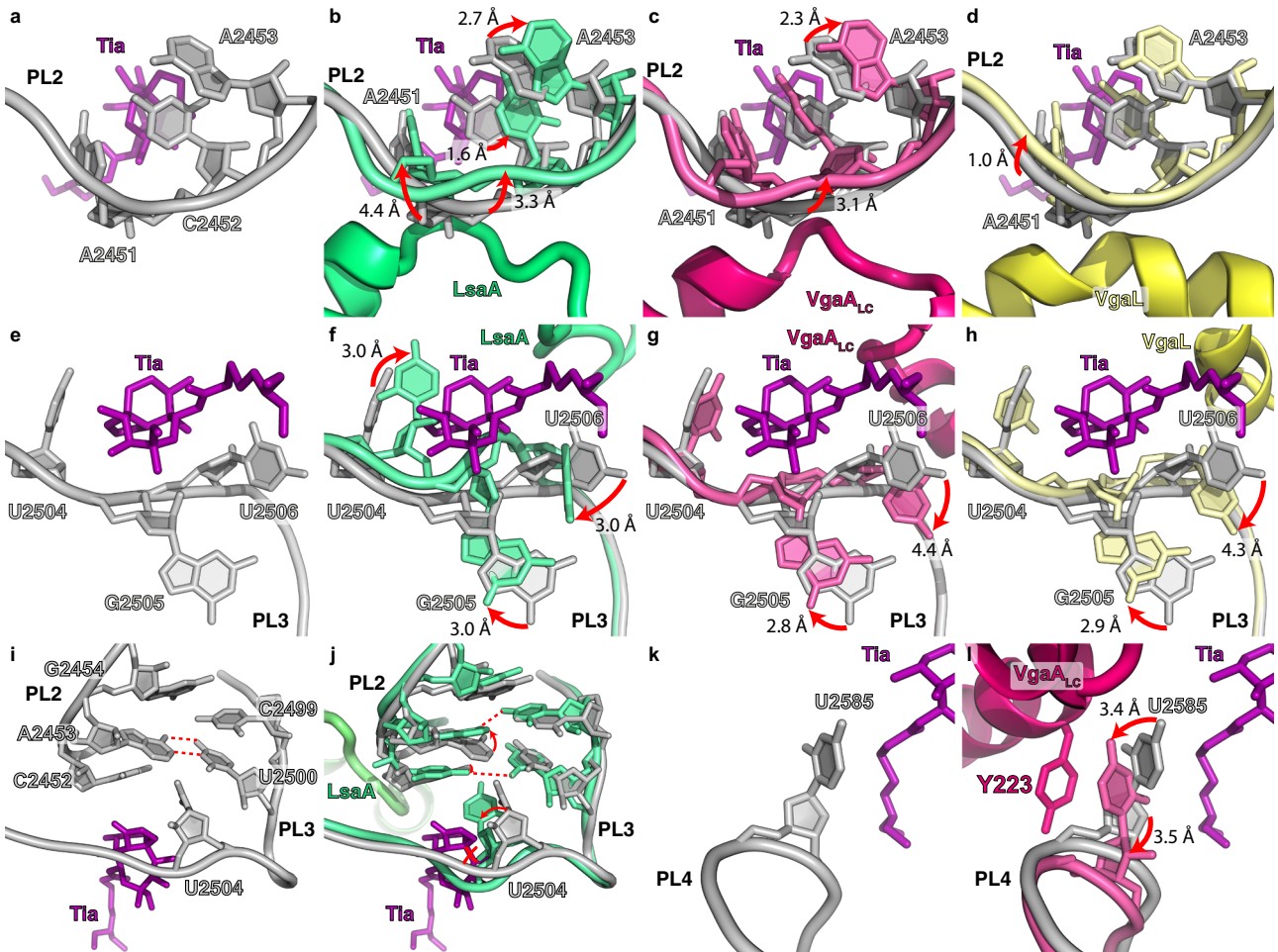

**Fig. 6 Changes in the PTC induced by ARE-ABCF binding. a–d** Effects of ARE binding on PL2 with respect to the tiamulin-binding site (PDB 1XBP)[2]. **a** The tiamulin-binding site only. **b–d** Same as **a** but with the LsaA- (**b**), VgaA_{LC}- (**c**), or VgaL-bound structure (**d**) superimposed. **e–h** Same as **a–d** but focused on PL3. **i**, **j** U2585 in the tiamulin site without (**i**) or with (**j**) VgaA_{LC} superimposed. Tyrosine223 of VgaA_{LC} is indicated. **k**, **l** Interaction between PL2 and PL3 contributing to the tiamulin-binding site, either without (**k**) or with (**l**) LsaA superimposed.

lincomycin (Supplementary Fig. 17). While the density for U2585 is not well-resolved in the LsaA- and VgaL-bound 70S structures, it appears nevertheless to adopt distinct conformations in the ARE-ABCFs compared to the drug-bound structures (Supplementary Fig. 17). By contrast, U2585 is clearly ordered in the VgaA_{LC}-70S structure where it stacks upon Tyr223 of VgaA_{LC} (Fig. 6l) in a position that precludes interaction with tiamulin (Fig. 6k, l) or virginiamycin M (Supplementary Fig. 17). Substituting Tyr223 of VgaA_{LC} to alanine diminished antibiotic resistance (Supplementary Table 2), indicating that the repositioning of U2585 is likely to contribute to antibiotic resistance conferred by this ARE-ABCF.

## Discussion

**Model of antibiotic resistance mediated by LsaA, VgaA_{LC}, and VgaL.** Based on our findings and the available literature on ARE-ABCFs, we propose a model for how the ARE-ABCFs LsaA, VgaA_{LC} and VgaL confer antibiotic resistance to their respective host organism (Fig. 7). PLS_A antibiotics have binding sites overlapping with the nascent polypeptide chain, and inhibit translation at, or soon after, initiation (Fig. 7a)[8–10]. As observed in our and previously reported structures[38,45], the incoming ARE-ABCFs bind in the E-site, triggering closure of the L1 stalk and inducing a distorted conformation of the P-tRNA. The ARD

extends into the antibiotic-binding pocket at the PTC causing drug release. In LsaA and VgaA_{LC}, the changes to the drug-binding site are substantial, while for VgaL the changes are rather subtle, as observed in other instances of antibiotic resistance[64,65] (Fig. 7b). We observed subpopulations of ARE-ABCF-bound complexes containing A-tRNA, suggesting that an incoming ternary complex can still be delivered to the A-site, despite the distortion of the P-tRNA (Fig. 7c). However, we note that our complexes were stalled with EQ_2-variant AREs, and in a natural context the ARE may bind and dissociate prior to an A-tRNA accommodation attempt. We propose that upon dissociation of the ARE-ABCF from the ribosome, the 3′ end of the A- and P-tRNAs can re-accommodate at the PTC (Fig. 7d). The trigger for nucleotide hydrolysis and exit of the ARE-ABCF from the E-site is unknown. In our model, rapid peptidyl transfer then creates a short nascent chain that overlaps with the antibiotic-binding site, thus preventing re-binding of the PLS_A drug until the next round of translation (Fig. 7d). We cannot exclude the possibility that an A-tRNA may also partially accommodate on the stalled initiation complex prior to ARE-ABCF binding, and become distorted as part of a 'knock-on' effect of P-tRNA disruption, consistent with the ability of ARE-ABCFs to 'reset' the P-tRNA independently of additional accommodation events[56]. In this model, potentially only one round of ATP hydrolysis per translation cycle is

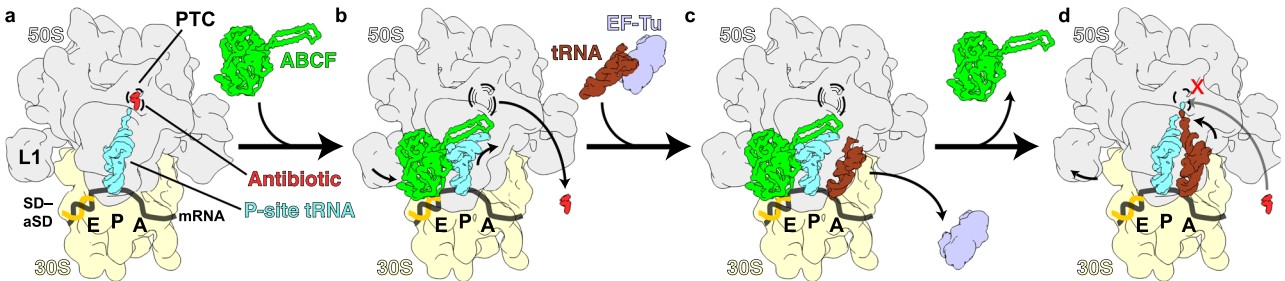

**Fig. 7 Model for ribosome protection by ARE-ABCFs VmlR, LsaA, VgaA_LC and VgaL. a** PLS_A-stalled ribosomes containing an initiator tRNA in the P-site are recognized by the ARE-ABCFs such as VmlR, LsaA, VgaA_LC and VgaL, which bind to the E-site of the ribosome with a closed ATP-bound conformation. **b** Binding of the ARE-ABCF induces a shifted P-site tRNA conformation in the ribosome allowing the ARD of the ARE-ABCF to access the peptidyl transferase centre (PTC). The ARD induces conformational changes within the 23S rRNA at the PTC that promotes dissociation of the drug from its binding site (shown as dashed lines). **c** Aminoacyl-tRNAs can still bind to the ARE-ABCF-bound ribosomal complex, but cannot accommodate at the PTC due to the presence of the ABCF and shifted P-site tRNA conformation. **d** Hydrolysis of ATP to ADP leads to dissociation of ARE-ABCF from the ribosome, which may allow the peptidyl-tRNA as well as the incoming aminoacyl-tRNA to simultaneously accommodate at the PTC. Peptide bond formation can then ensue, converting the ribosome from an initiation to an elongation (pre-translocation) state, which is resistant to the action of initiation inhibitors, such as PLS_A antibiotics.

necessary to confer resistance. We can also not exclude that the P-tRNA dissociates following release of the ARE-ABCF and/or that other factors are involved in recycling of the post-antibiotic release complexes.

ARE-ABCFs such as LsaA, VgaA_LC, VgaL and VmlR confer resistance to PLS_A antibiotics but not phenicols or oxazolidinones[25]. This observation has been puzzling, as both groups of antibiotics have overlapping binding sites[2–6]. However, phenicols and oxazolidinones inhibit translation during elongation at specific motifs[9,66], while PLS_A antibiotics instead inhibit translation at the initiation stage[8–10]. This suggests that ARE-ABCFs such as LsaA, VgaA_LC, VgaL and VmlR are likely to be specific for initiation complexes, whereas ARE-ABCFs such as OptrA and PoxtA may have an additional specificity for drug-stalled elongation complexes. It will be interesting in the future to see how OptrA and PoxtA remove phenicols and oxazolidinones from the ribosome given the short ARD is not predicted to be able to reach into the PTC.

Another question is whether the EQ_2-substituted ATPase-deficient variants of ARE-ABCF, like the ones used in this study, bind the ribosome in the pre- or post-antibiotic-release state (Fig. 7b). Although direct evidence is lacking, three reasons lead us to propose that these proteins are bound in the post-antibiotic-release state:

1. In the case of LsaA, VgaA_LC and VmlR the position of the ARD directly overlaps with the antibiotic-binding site. Although the side chain of the overlapping amino acid is not critical for antibiotic resistance in most instances, the overlap nonetheless implies mutually exclusive binding.
2. MsrE-EQ_2 stimulates dissociation of azithromycin from the ribosome[38].
3. Our attempts to form complexes containing both antibiotic and ARE-ABCF have been unsuccessful, resulting in exclusive binding of either the ARE-ABCF or the antibiotic, similarly to what we observed for TetM, a tetracycline-resistance ribosome protection protein[67].

How does the ARE-ABCF ARD mediate antibiotic resistance (Fig. 7b, c)? In one model, by analogy to the TetM tetracycline-resistance protein[11,68], the ARD may induce antibiotic dissociation by a direct steric overlap with the antibiotic. In the case of VmlR, substitutions of the Phe237 residue that overlaps the binding site of PLS_A antibiotics affect resistance to one of three relevant antibiotics, indicating that both direct steric overlap and an indirect mechanism—for example, modulation of the

antibiotic-binding site—can contribute to resistance[45]. In the case of MsrE substitution of Leu242, which overlaps with the erythromycin binding site, as well as adjacent residues abolished or severely reduced the antibiotic resistance activity of this protein[38]. In both cases, a mixture of direct steric overlap and indirect long-distance effects is consistent with the available data[24]. The ARDs of LsaA, VgaA_LC and VgaL either do not directly overlap with the PLS_A-binding site, or where there is an overlap, as with LsaA Phe257 and VgaA_LC Val219, the side chains are not essential for resistance, implicating an indirect mechanism for these proteins (Figs. 4–6, Supplementary Figs. 14–16 and Supplementary Table 2). Alanine mutagenesis instead indicates that the side chains of residues surrounding the amino acid closest to the antibiotic-binding pocket, as well as those that contact the 23S rRNA, are necessary for resistance (Fig. 5c, Supplementary Figs. 11 and 12 and Supplementary Table 2). These residues may position the ARD in the PTC. No single set of 23S rRNA rearrangements was identical among LsaA, VgaA_LC and VgaL, although displacement of PTC loops PL2 and PL3, especially residue U2504, was ultimately observed in each ARE-ABCF-70S structure (Fig. 6). Broadly, changes to the PTC were similar between the VgaA_LC- and VgaL-bound 70S structures (Fig. 5g–i and Supplementary Figs. 14–16), consistent with the grouping of these proteins together in the ARE1 subfamily[20]. While structures of the same or similar antibiotic bound to ribosomes from different species are generally similar, we cannot completely exclude that some differences in nucleotide conformations arise because of comparing our ARE-ABCF-bound PTC conformations with antibiotic-ribosome structures from different species, for example, E. coli for VgM[61] and D. radiodurans for tiamulin[2]. Similarly, some conformational variability can also arise due to limitations in resolution of some of the antibiotic structures, such as the tiamulin-50S structure that was reported at 3.5 Å[2] and the S. aureus lincomycin-50S structure at 3.7 Å[6]. A future goal could be to determine higher resolution structures of the antibiotic-stalled ribosomal complexes prior to ARE-ABCF binding and from the same organisms as the ARE-ABCF.

In summary, we present three structures of ARE-ABCFs bound to 70S ribosomes from relevant Gram-positive pathogenic bacteria and present the model of the ribosome from *Listeria monocytogenes*. Our structures and mutagenesis experiments support an indirect mechanism of ARE-ABCF action, in which a conformational selection in the PTC, elicited by ARE binding to the ribosome, leads to antibiotic egress, and hint at a

rationalization for the specificity of LsaA, VgaA$_{LC}$ and VgaL for PLS$_A$ antibiotics. Each ARE-ABCF binds the 70S similarly as observed for other bacterial ABCF proteins, but alters the geometry of the PTC distinctively, consistent with the convergent evolution—and divergent sequences—of this class of ABCF proteins.

## Methods

**Strains and plasmids**. All strains and plasmids used in this work are listed in Table S5. Primers are listed in Table S6.

*E. faecalis*. OG1RF and TX5332, a LsaA disruption mutant of OG1RF[30], were kindly provided by Dr. Barbara E. Murray (Health Science Center, University of Texas). All cloning was performed by Protein Expertise Platform at Umeå University. *E. faecalis* LsaA ORF was PCR amplified from pTEX5333 plasmid and cloned into pCIE vector[69] for cCF10-induced expression. The LsaA ORF was supplemented with C-terminal His$_6$-TEV-FLAG$_3$-tag (HTF tag) and the ribosome-binding site was optimized for high expression yield. Point mutations E$_{142}$Q and E$_{452}$Q were introduced to LsaA resulting in pCIE_LsaA-EQ$_2$-HTF.

*S. haemolyticus*. vga(A)$_{LC}$ gene was PCR-amplified from a *S. haemolyticus* isolate held in the O'Neill strain collection at the University of Leeds, using oligonucleotide primers vgaA$_{LC}$-F (5′-GGTGGTGGTACCAGGATGAGGAAAATATGA AAA-3′) and vgaA$_{LC}$-R (5′-GGTGGTGAATTCAGGTAATTTATTTATCTAAA TTTCTT-3′) (engineered restriction sites shown underlined). The protein encoded by this gene is identical to that previously reported[50] (accession number DQ823382). The fragment was digested with *Kpn*I and *Eco*RI and ligated into the tetracycline-inducible expression vector pRMC2 (ref. [70]). Constructs encoding the VgaA$_{LC}$ protein fused with a C-terminal FLAG$_3$ tag were obtained by synthesis (Genewiz), with E$_{105}$Q, E$_{410}$Q and EQ$_2$ mutants subsequently created by site-directed mutagenesis. Generation of other point mutants of untagged Vga(A)$_{LC}$ was performed by NBS Biologicals, again using chemical synthesis to generate the original vga(A)$_{LC}$ template, followed by site-directed mutagenesis.

*L. monocytogenes*. VgaL (Lmo0919). In order to construct *L. monocytogenes* EGDe::Δ*lmo0919*, regions corresponding to the upstream and downstream flanking regions of *lmo0919*, present on the EGDe genome were amplified with primer pairs VKT35 (5′-GGGGGGATCCATCACTAGCCGAATCCAAAC-3′), VKT36 (5′-ggg ggaattcaaaaaataacctcctgaatattttcagag-3′) and VHKT37 (5′-GGGGGAATTCAAAA AATAACCTCCTGAATATTTTCAGAG-3′), VHKT38 (5′-GGGGCCATGGCG TGCTGTACGGTATGC-3′), respectively. Fragments were then cloned in tandem into the pMAD vector using *Bam*HI, *Eco*RI and *Nco*RI restriction sites. The resulting vector, VHp689, was then sequenced to ensure wild-type sequences of clones. Gene deletion was then performed as per Arnaud et al.[71].

*lmo0919* was amplified from EGDe genomic DNA using primers VHKT12 (5′-CCCCCCATGGCATCTACAATCGAAATAAATC-3′) and VHKT39 (5′-GGGGGCTGCAGTTAACTAAATTGCTGTCTTTTTG-3′), and cloned into pIMK3 using *Nco*I and *Pst*I restriction sites, resulting in plasmid VHp690.

Overlap extension PCR was used in order to introduce a HTF tag at the C-terminus of *lmo0919* (ref. [72]). The *lmo0919* locus and HTF tag were amplified with primer pairs VHKT12, VHKT15 (5′-ATGATGATGGCCGCCACTAAATTGCT GTCTTTTTG-3′) and VHKT14 (5′-AGACAGCAATTTAGTGGCGGCCATC ATCATCATC-3′), VHKT13 (5′-GGGGGCTGCAGTTAGCCTTTGTCATCGTC-3′) using EGDe genomic DNA and VHp100 template DNA, respectively, producing fragments with overlapping ends. VHKT12 and VHKT13 were then used to fuse the fragments and the resulting PCR product was cloned into pIMK3 using *Nco*I and *Pst*I sites resulting in VHp692.

To introduce two EQ mutations (E104Q and E408Q) simultaneously into the VHp692 plasmid, primers VHT266 (5′-TCTTGATCAACCAACCAACTATTTGG ATATCTACGCAATGGAA-3′) and VHT267 (5′-TTGTTGGTTGGTCTGCTAG GAGAACACTTGGATTTTGGCGCA-3′) containing both mutations were used to extend out from *lmo0919*$^{HTF}$ to amplify the VHp692 backbone. Primers VHT264 (5′-AGCAGACCAACCAACAAGCAATCTTGATGTCG-3′) and VHT265 (5′-TG GTTGGTTGATCAAGAATCAAGAAATTGGCGT-3′) also containing *lmo0919*$^{EQ2}$ mutations were used to amplify a fragment with overlapping sequence to the backbone fragment. Both PCR products were then assembled using NEBuilder® HiFi DNA Assembly Master Mix (NEB), resulting in VHp693.

*B. subtilis*. To construct the VHB109 [*trpC2 ΔvmlR* thrC::P$_{hy-spnak}$-*lsaA kmR*] strain untagged LsaA under the control of an IPTG-inducible P$_{hy-spank}$ promotor, a PCR product encoding lsa(A) was PCR-amplified from pTEX5333 using the primers VHT127 (5′-CGACGAAGGAGAGAGGCGATAATGTCGAAAATTGAACTAA AACAACTATC-3′) and VHT128 (5′-CACCGAATTAGCTTGCATGCTTATGA TTTCAAGACAATTTTTTTATCTGTTA-3′). The second PCR fragment encoding a kanamycin-resistance marker, a polylinker downstream of the Phy-spank promoter and the lac repressor ORF—all inserted in the middle of the thrC gene—was PCR-amplified from pHT009 plasmid using primers VHT123 (5′-CATTATC GCTCTCTCCTTCGTCGACTAAGCTAATTG-3′) and VHT125 (5′-TAAGCA

TGCAAGCTAATTCGGTGGAAACGAGG-3′). The two fragments were ligated using the NEBuilder HiFi DNA Assembly master mix (New England BioLabs, Ipswich, MA) yielding the pHT009-lsaA plasmid (VHp369) which was used to transform the VHB5 [*trpC2 ΔvmlR*] strain. Selection for kanamycin resistance yielded the desired VHB109 strain. To construct the VHB168 [*trpC2 ΔvmlR* thrC:: P$_{hy-spnak}$-*lsaAK244A kmR*] strain, VHp369 plasmid was subjected to site-directed mutagenesis using primer VHP303 (5′-GCATCACCTTCACGGTTCATCGACC ATTCCGCT-3′) and VHP304 (5′-GTACGGCAACGCTAAGGAAAAAGGGA GCGGGGCGA-3′), according to the directions of Phusion Site-Directed Mutagenesis Kit (Thermo Fisher Scientific), yielding VHp526 (pHT009-*lsaAK244A*) plasmid which was used to transform the VHB5 [*trpC2 ΔvmlR*] strain. Selection for kanamycin resistance yielded the desired VHB168 strain. To construct the VHB169 [*trpC2 ΔvmlR* thrC::P$_{hy-spnak}$-*lsaAF257A kmR*] strain, VHp369 plasmid was subjected to site-directed mutagenesis using primer VHP305 (5′-CAATCGCCCCGC TCCCTTTTTCCTTAGCGT-3′) and VHP306 (5′-CGGATACAGGAGCCATT GGTGCCCGGGCA-3′), according to the directions of Phusion Site-Directed Mutagenesis Kit (Thermo Fisher Scientific), yielding, yielding VHp527 (pHT009-*lsaAF257A*) plasmid which was used to transform the VHB5 [*trpC2 ΔvmlR*] strain. Selection for kanamycin resistance yielded the desired VHB169 strain.

**Bacterial transformation**

*E. faecalis*. Electrocompetent cells were prepared as per Bhardwaj et al.[73]. In short, an overnight culture grown in the presence of appropriate antibiotics was diluted to OD$_{600}$ of 0.05 in 50 mL of BHI media (supplemented with 2 mg/mL kanamycin in case of TX5332), grown to OD$_{600}$ of 0.6–0.7 at 37 °C with moderate shaking (160 r. p.m.). Cells were collected by centrifugation at 3200 × g at 4 °C for 10 min. Cells were resuspended in 0.5 mL of sterile lysozyme buffer (10 mM Tris-HCl pH 8; 50 mM NaCl, 10 mM EDTA, 35 µg/mL lysozyme), transferred to 1.5 mL Eppendorf tube and incubated at 37 °C for 30 min. Cells were pelleted at 8700 × g at 4 °C for 10 min and washed three times with 1.5 mL of ice-cold electroporation buffer (0.5 M sucrose, 10% glycerol(w/v)). After last wash the cells were resuspended in 500 µL of ice-cold electroporation buffer and aliquoted and stored at –80 °C. For electroporation 35 µL of electrocompetent cells were supplemented with 1 µg of plasmid DNA, transferred to ice-cold 1 mm electroporation cuvette and electroporated at 1.8 keV. Immediately after electroporation 1 mL of ice-cold BHI was added to the cells, the content of the cuvette was transferred to 1.5 mL Eppendorf tubes and the cells were recovered at 37 °C for 2.5 h and plated onto BHI plates containing appropriate antibiotics (10 µg/mL chloramphenicol and 2 mg/mL kanamycin).

*S. aureus*. Preparation and transformation of *S. aureus* electrocompetent cells followed the method of Schenk and Laddaga[74], though used TSBY (Tryptone soya broth [Oxoid] containing 2.5% yeast extract) in place of B2 medium. Briefly, bacteria were grown with vigorous aeration in TSBY to an OD$_{600}$ of 0.6, harvested by centrifugation, and washed three times in an equal volume of sterile, deionized water. Subsequent wash steps used decreasing volumes of 10% glycerol; first 1/5 the original culture volume, then 1/10, finally resuspending in ~1/32 volume and storing the resultant electrocompetent cells at −80 °C. For electroporation, 60 µL of electrocompetent cells were mixed with ≥1 µg of plasmid DNA in a 1 mm electroporation cuvette at room temperature and pulsed at 2.3 kV, 100 Ω, 25 µFD. Immediately after electroporation, 390 µL room temperature TSBY was added to the cells and incubated with aeration at 37 °C for 1–2 h, before plating onto tryptone soya agar with appropriate antibiotic selection. Using this method, sequence-verified constructs established in *E. coli* were first transferred into the restriction deficient *S. aureus* RN4220 strain[75], before recovery and introduction into *S. aureus* SH1000 (refs. [76,77]).

*L. monocytogenes*. *L. monocytogenes* EGD-e was transformed with pIMK3 integrative plasmids via conjugation. *E. coli* S17.1 harbouring pIMK3 and its derivatives was grown at 37 °C overnight in LB media supplemented with 50 µg/mL kanamycin; 1 mL of culture was washed three times with sterile BHI media to remove antibiotics. Two hundred microliters of washed *E. coli* culture was mixed with an equal volume of *L. monocytogenes* overnight culture grown at 37 °C in BHI media. Two hundred microliters of mixed bacterial suspension was then dropped onto a conjugation filter (Millipore #HAEP047S0) placed onto a BHI agar plate containing 0.2 µg/mL penicillin-G. After overnight incubation at 37 °C, bacterial growth from the filter was resuspended in 1 mL of BHI and 100–300 µL plated onto BHI agar plates supplemented with 50 µg/mL kanamycin (to select for pIMK3), 50 µg/mL nalidixic acid and 10 µg/mL colistin sulfate (Sigma-Aldrich C4461-100MG). Resulting colonies were checked for correct integration via PCR and subsequent sequencing using primers VHKT42 and VHKT43.

**Antibiotic susceptibility testing**. Minimum inhibitory concentrations (MIC) were determined based on guidelines from the European Committee on Antimicrobial Susceptibility Testing (EUCAST) (http://www.eucast.org/ast_of_bacteria/mic_determination).

*E. faecalis*. Bacteria were grown in BHI media supplemented with 2 mg/mL kanamycin (to prevent *lsa* revertants), 0.1 mg/mL spectinomycin (to maintain the pCIE$_{spec}$ plasmid), 100 ng/mL of cCF10 peptide (to induce expression of LsaA

protein) as well as increasing concentrations of antibiotics was inoculated with $5 \times 10^5$ CFU/mL ($OD_{600}$ of approximately 0.0005) of *E. faecalis* $\Delta lsaA$ (*lsa::Kan*) strain TX5332 transformed either with empty $pCIE_{spec}$ plasmid or with $pCIE_{spec}$ encoding LsaA. After 16–20 h at 37 °C without shaking, the presence or absence of bacterial growth was scored by eye.

*S. aureus*. Bacteria were grown in cation-adjusted Mueller-Hinton Broth (MHB) at 37 °C with vigorous aeration, supplemented with 10 mg/L chloramphenicol to maintain the pRMC2 plasmid. Upon reaching an absorbance of $OD_{625}$ of 0.6, anhydrotetracycline (ATC) (Sigma-Aldrich, UK) was added at a final concentration of 100 ng/mL to induce expression from pRMC2, and incubated for a further 3 h. Cultures were then diluted to $5 \times 10^5$ CFU/mL using MHB supplemented with ATC (100 ng/mL) and used in MIC determinations essentially as described above (though cultures were shaken).

*L. monocytogenes*. Bacteria were grown in BHI media supplemented with 50 µg/mL kanamycin (to prevent loss of the integrated pIMK3 plasmid), 1 mM of IPTG (to induce expression of VgaL protein) as well as increasing concentrations of antibiotics was inoculated with $5 \times 10^5$ CFU/mL ($OD_{600}$ of approximately 0.0003) of *L. monocytogenes* EGD-e wild-type strain or EGD-e::$\Delta lmo0919$ strain transformed either with empty pIMK3 plasmid or with pIMK3 encoding VgaL variants. After 16–20 h at 37 °C without shaking, the presence or absence of bacterial growth was scored by eye.

*B. subtilis (for LsaA mutants)*. *B. subtilis* strains were pre-grown on LB plates supplemented with 1 mM IPTG overnight at 30 °C. Fresh individual colonies were used to inoculate filtered LB medium in the presence of 1 mM IPTG, and $OD_{600}$ adjusted to 0.01. The cultures were seeded on a 100-well honeycomb plate (Oy Growth Curves AB Ltd, Helsinki, Finland), and plates incubated in a Bioscreen C (Labsystems, Helsinki, Finland) at 37 °C with continuous medium shaking. After 90 min ($OD_{600} \approx 0.1$), antibiotics were added and growth was followed for an additional 6 h.

### Preparation of bacterial lysates
*Preparation of bacterial biomass. E. faecalis*: *E. faecalis* TX5332 transformed with pCIE plasmids (either empty vector and expressing either wild type or $EQ_2$ variants of C-terminally HTF-tagged LsaA) were grown overnight from single colony in BHI supplemented with 2 mg/mL kanamycin and 10 µg/mL of chloramphenicol. Next day overnight cultures were diluted to starting $OD_{600}$ of 0.05 in 160 mL BHI supplemented with 0.5 mg/mL kanamycin and 10 µg/mL of chloramphenicol. Cells were grown with intensive shaking at 37 °C till $OD_{600}$ of 0.6 and were induced with 300 ng/mL of cCF10 peptide for 30 min prior harvesting by centrifugation at $10,000 \times g$ 15 min at 4 °C.

*S. aureus*: *S. aureus* SH1000 transformed with pRMC2 plasmids (empty vector, wild type and $EQ_2$ VgaA$_{LC}$-FLAG$_3$) were grown in LB supplemented with 25 µg/mL of chloramphenicol. Saturated cultures were diluted to an $OD_{600}$ of 0.1 in 400 mL LB supplemented with 20 µg/mL of chloramphenicol and grown at 37 °C with vigorous aeration to an $OD_{600}$ of 0.6. Protein expression was induced with 100 ng/mL of anhydrotetracycline for 30 min prior to harvesting by centrifugation at $10\,000 \times g$ for 15 min at 4 °C.

*L. monocytogenes*: *L. monocytogenes* EGD-e was transformed with pIMK3 plasmids (empty vector, wild type and $EQ_2$ VgaL-HTF) were grown overnight from single colony in LB supplemented with 50 µg/mL of kanamycin. Next day overnight cultures were diluted till starting $OD_{600}$ of 0.005 in 200 mL BHI supplemented with 50 µg/mL of kanamycin. Cells were grown at 37 °C with shaking at 160 r.p.m. till $OD_{600}$ of 0.6 and were induced with 1 mM IPTG for 60 min prior harvesting by centrifugation at $10,000 \times g$ for 15 min at 4 °C.

*Preparation of clarified lysates*. Cell pellets were resuspended in 1.5 mL of cell lysis buffer (95 mM KCl, 5 mM NH$_4$Cl, 20 mM HEPES pH 7.5, 1 mM DTT, 5 mM Mg (OAc)$_2$, 0.5 mM CaCl$_2$, 8 mM putrescine, 1 mM spermidine, 1 tablet of cOmplete™ EDTA-free Protease Inhibitor Cocktail (Roche) per 10 mL of buffer and in the absence or presence of either 0.5 or 0.75 mM ATP), resuspended cells were opened by a FastPrep homogeniser (MP Biomedicals) with 0.1 mm zirconium beads (Techtum) in four cycles by 20 s with 1 min chill on ice. Cell debris was removed after centrifugation at $14,800 \times g$ for 15 min at 4 °C. Total protein concentration in supernatant was measured by Bradford assay (Bio-Rad), supernatant was aliquoted and frozen in liquid nitrogen.

### Polysome fractionation and immunoblotting
*Sucrose density gradient centrifugation*. After melting the frozen lysates on ice, 2 A$_{260}$ units of each extract was aliquoted into three tubes and supplemented with or without 0.5–0.75 mM ATP and was loaded onto 5–25 or 7–35% (w/v) sucrose density gradients in HEPES:Polymix buffer[78], 5 mM Mg(OAc)$_2$ and supplemented or not with 0.5–0.75 mM ATP. Gradients were resolved at $245,000 \times g$ for 2.5 h at 4 °C in an SW41 rotor (Beckman) and analysed and fractionated using Biocomp Gradient Station (BioComp Instruments) with A$_{280}$ as a readout.

*Immunoblotting. LsaA and VgaA$_{LC}$*: Schleicher & Schuell Minifold II Slot Blot System SRC072/0 44-27570 manifold was used for transferring samples from sucrose gradient fractions to PVDF membranes (Immobilon PSQ, Merck Millipore). Shortly, 15–100 µL of each sucrose gradient fraction was added to 200 µL of slot-blotting buffer (20 mM HEPES:KOH pH 7.5, 95 mM KCl, 5 mM NH$_4$Cl, 5 mM Mg(OAc)$_2$) in slots and blotted onto PVDF membrane that had been activated with methanol for 1 min, wetted in MilliQ water and equilibrated with Slot-blotting Buffer (1c PM 5 mM Mg$^{2+}$ without putrescine and spermidine) for 10 min. After blotting of the samples each slot was washed twice with 200 µL of Slot-blotting Buffer. The membrane was removed from the blotter, transferred to hybridization bottle, equilibrated for 10 min in PBS-T (1× PBS supplemented with 0.05% Tween-20) and blocked in PBS-T supplemented with 5% w/v nonfat dry milk for 1 h. Antibody incubations were performed for 1 h in 1% nonfat dry milk in PBS-T with five 5-min washes in fresh PBS-T between and after antibody incubations. HTF-tagged LsaA and FLAG$_3$-tagged VgaA$_{LC}$ proteins were detected using anti-Flag M2 primary (Sigma-Aldrich, F1804; 1:10,000 dilution) antibodies combined with anti-mouse-HRP secondary (Rockland; 610-103-040; 1:10,000 dilution) antibodies. An ECL detection was performed on ImageQuant LAS 4000 (GE Healthcare) imaging system using Pierce® ECL western blotting substrate (Thermo Scientific). The blotting and all incubations were performed at room temperature in a hybridization oven.

*VgaL (Lmo0919)*: Western blotting of lysates on sucrose gradient fractionation was performed as previously described[78]. In all, 1.5 mL of 99.5% ethanol was added to each 0.5 mL sucrose fraction and precipitated at −20 °C overnight. Samples were then pelleted via centrifugation for 30 min at $14,800 \times g$, air dried and resuspended in 2× SDS loading buffer (100 mM Tris-HCl pH 6.8, 4% SDS w/v, 0.02% bromophenol blue, 20% glycerol (w/v), 4% β-mercaptoethanol). Samples were resolved on a 12% SDS-PAGE gel and transferred to a nitrocellulose membrane (pore size 0.2 µM, BioTrace™ NT) using the Bio-Rad Trans-Blot Turbo Transfer apparatus (30 min, 1 A, 25 V). The membrane was then blocked for 1 h at room temperature in PBS-T (1× PBS, 0.05% Tween-20) with 5% (w/v) nonfat dry milk. VgaL-HTF was then detected using anti-Flag M2 primary antibodies as described above for Lsa and VgaA$_{LC}$. VgaL-HTF was detected using anti-Flag M2 primary (Sigma-Aldrich, F1804; 1:10,000 dilution) antibodies combined with anti-mouse-HRP secondary (Rockland; 610-103-040; 1:10,000 dilution) antibodies.

### Affinity purification on anti-FLAG M2 affinity gel
One hundred microlitres of well mixed anti-FLAG M2 Affinity Gel aliquots were loaded on columns (Micro Bio-Spin Columns, Bio-Rad) and washed two times with 1 mL of cell lysis buffer by gravity flow. All incubations, washings and elutions were done at 4 °C.

The total protein concentration of each lysate was adjusted to 2 mg/mL with cell lysis buffer and 1 mL of each lysate was loaded on columns and incubated for 2 h with end-over-end mixing for binding. The columns were washed five times by 1 mL of cell lysis buffer by gravity flow. For elution of FLAG-tagged proteins and their complexes 100–300 µL of 0.1 mg/mL FLAG$_3$ peptide (Sigma) was added to samples, the solutions were incubated at 4 °C for 20 min with end-over-end mixing. Elutions were collected by centrifugation at $2000 \times g$ for 2 min at 4 °C.

Twenty microlitre-aliquots of collected samples (flow-through, washes and elutions) were mixed with 5 µL of 5× SDS loading buffer and heated up at 95 °C for 15 min. The beads remaining in the column were washed twice with 1 mL of cell lysis buffer and resuspended in 100 µL of 1× SDS loading buffer. Denatured samples were resolved on 12–15% SDS-PAGE. SDS-gels were stained by Blue-Silver Coomassie Staining[79] and washed with water for 6 h or overnight before imaging with LAS 4000 (GE Healthcare).

### tRNA microarrays
To fully deacylate tRNAs, eluates and input lysate samples from two biological replicates were mixed with 80 µL 250 mM Tris-HCl, pH 9.0, 10 µL 0.2 M EDTA, 10 µL 1% SDS, and incubated for 45 min, and neutralized with 200 µL 1 M NaOAc, pH 5.5, before mixing 1:1 with acidic phenol:chloroform alcohol 5:1. The supernatant was precipitated with ethanol and dissolved in ddH$_2$O.

tRNA microarrays were performed as described[80]. Briefly, using the unique invariant single-stranded 3′-NCCA-ends of intact tRNA a Cy3-labelled or Atto647-labelled RNA/DNA hybrid oligonucleotide was ligated to the tRNA extracted from the ARE-immunoprecipitated samples and total *E. faecalis* tRNA (from the lysate), respectively. Labelled tRNA was purified by phenol:chloroform extraction and loaded on a microarray containing 24 replicates of full-length tDNA probes recognizing *E. faecalis* tRNA isoacceptors. Fluorescence signals were normalized to four in vitro-transcribed human tRNAs, spiked into each sample. Microarrays were statistically analysed with in-house scripts written in Python 3.7.0. Data are available at the Gene Expression Omnibus under accession GSE 162168.

### Grid preparation, cryo-electron microscopy and single-particle reconstruction
*Preparation of cryo-EM grids and data collection*. Elutions from LsaA and VgaL pull-downs were loaded on grids within 2 h after obtaining them without freezing, samples were kept on ice. The VgaA$_{LC}$ sample was frozen in liquid nitrogen after pull-down, defrosted and loaded later. After glow-discharging of grids, 3.5 µL of sample was loaded on grids in Vitrobot (FEI) in conditions of 100% humidity at 4 °C, blotted for 5 s and vitrified by plunge-freezing in liquid ethane. Samples were

imaged on a Titan Krios (FEI) operated at 300 kV at a nominal magnification of ×130k (LsaA) or ×165k (VgaA$_{LC}$ and VgaL, 1.09 Å/pixel and 0.86 Å/pixel, respectively, later estimated to be 1.041 and 0.82 Å/pixel, respectively, by comparing refined maps to structures with known magnification) with a Gatan K2 Summit camera at an exposure rate of 5.80 electrons/pixel/s with a 4 s exposure and 40 frames (LsaA), or 20 frames (VgaA$_{LC}$ and VgaL) using the EPU software. Quantifoil 1.2/1.3 Cu$_{200}$ grids were used for LsaA and VgaA$_{LC}$ and Quantifoil 2/2 Cu$_{200}$ grids were used for VgaL.

*Single-particle reconstruction.* Motion correction was performed with MotionCor2 with 5 × 5 patches[81]. Relion 3.0 or 3.1 was used for further processing unless otherwise stated and resolutions are reported according to the so-called 'gold standard' criteria[82–84]. CTFFIND4 (LsaA dataset) or Gctf v1.06 (VgaA$_{LC}$ and VgaL datasets) was used for CTF estimation[85,86]. Particles were picked with Gautomatch (https://www2.mrc-lmb.cam.ac.uk/research/locally-developed-software/zhang-software/#gauto, developed by K. Zhang) without supplying a reference, and in the case of LsaA, re-picked using RELION autopicker after templates were generated by 2D classification. Particles were initially extracted at three times the original pixel size and subjected to 2D classification. Classes that resembled ribosomes were used for 3D refinement, with a 60 Å low-pass filter applied to initial references. For 3D refinement of LsaA-70S, the initial reference was EMDB-0176, a *B. subtilis* 70S ribosome with no factor bound in the E-site[45]; for VgaA$_{LC}$-70S and VgaL-70S 3D refinements the RELION initial model job type was used to create a reference from particles selected after 2D classification. 3D classification was performed without angular sampling, and classes of interest were re-extracted at 1.041 Å/pixel (LsaA) or 0.82 Å/pixel (VgaA$_{LC}$ and VgaL) for further refinement.

In the case of LsaA, after initial 3D classification, a soft mask around the A-site was used for partial signal subtraction followed by focussed classification. The classes with the strongest and weakest A-site density were selected for signal restoration and refinement. In the case of the VgaA$_{LC}$ dataset, initial 3D classification yielded a class with apparent sub-stoichiometric density in the E-site corresponding to VgaA$_{LC}$. Micrographs with poor values from CTF estimation were discarded, particles were re-extracted, subjected to an additional 2D classification and 3D refinement, followed by Bayesian polishing and CTF refinement. An additional 3D classification yielded a class with strong E-site density corresponding to the factor. Refer to Supplementary Figs. 4–6 for details.

For multibody refinements, soft masks around the small subunit body, small subunit head, and large subunit/ARD were applied. In the case of the VgaA$_{LC}$ dataset, particles were first re-extracted in a smaller box (360 × 360 pixels) and subjected to 3D refinement prior to multibody refinement.

ResMap was used to estimate local resolution[87]. Maps were locally filtered using SPHIRE[88].

*Molecular modelling.* For the *E. faecalis* and *L. monocytogenes* ribosomes, homology models were generated with SWISS-MODEL[89], mostly from PDB 6HA1/6HA8 (ref. [45]). PDBs 4YBB[90] 5MDV[91] were used as additional templates and references where necessary, 4V9O[92] was used for bS21, 7K00 (ref. [93]) for bL31, 5ML7 (ref. [94]) and 3U4M[95] were used for the L1 stalk region, 5AFI[96] and 5UYQ[97] were used for tRNAs, and 6QNQ was used to help tentatively place metal ions[98]. PDB 5LI0 (ref. [57]) was used as a starting model for the *S. aureus* ribosome. Where appropriate, individual components of multibody refinements were fitted into density from the corresponding locally filtered map to help modelling. Models were adjusted with Coot[99] and refined using locally filtered maps in Phenix version dev-2947-000 (ref. [100]).

Figures were created with PyMOL 2.0 (Schrödinger, LLC), UCSF Chimera[101], UCSF ChimeraX[102], RELION[82], and Igor Pro (WaveMetrics, Inc.). Structures were aligned in PyMOL using the 23S rRNA unless otherwise noted. Subunit rotation was visualized in PyMOL using the modevectors script, which was initially developed by Sean Law and modified by others, and the rotation angle measured using the draw_axis script, made by Pablo Guardado Calvo.

Figures were assembled with Adobe Illustrator (Adobe Inc.).

**Reporting summary**. Further information on research design is available in the Nature Research Reporting Summary linked to this article.

## Data availability
Micrographs have been deposited as uncorrected frames in the Electron Microscopy Public Image Archive (EMPIAR) with the accession codes EMPIAR-10682 (LsaA immunoprecipitation), EMPIAR-10683 (VgaA$_{LC}$ immunoprecipitation), and EMPIAR-10684 (VgaL immunoprecipitation). Cryo-EM maps have been deposited in the Electron Microscopy Data Bank (EMDB) with accession codes EMD-12331 (LsaA-70S), EMD-12332 (VgaA$_{LC}$-70S), EMD-12333 (*S. aureus* 70S with P-tRNA from VgaA$_{LC}$ immunoprecipitation) and EMD-12334 (VgaL-70S). Molecular models have been deposited in the Protein Data Bank with accession codes 7NHK (LsaA-70S), 7NHL (VgaA$_{LC}$-70S), 7NHM (*S. aureus* 70S with P-tRNA from VgaA$_{LC}$ immunoprecipitation) and 7NHN (VgaL-70S). Microarray data have been deposited in Gene Expression Omnibus under accession GSE 162168. Scripts for analysing microarray data are available upon request to C.P. and Z.I. Source data are provided with this paper.

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

## Acknowledgements
We are grateful to Barbara E. Murray for sharing *E. faecalis* Δ*lsaA* (*lsa*::Kan) strain TX5332 (ref. [30]) and Jose A. Lemos for sharing the pCIE plasmid vector[69]. We thank Michael Hall for help with cryo-EM data collection. We would also like to thank the reviewers for their comments that improved the manuscript. The electron microscopy data were collected at the Umeå Core Facility for Electron Microscopy, a node of the Cryo-EM Swedish National Facility, funded by the Knut and Alice Wallenberg, Family Erling Persson and Kempe Foundations, SciLifeLab, Stockholm University and Umeå University. This work was supported by the Deutsche Forschungsgemeinschaft (DFG) (grant WI3285/8-1 to D.N.W.), the Swedish Research Council (Vetenskapsrådet) grants (2017-03783 to V.H. and 2019-01085 to G.C.A.), Ragnar Söderbergs Stiftelse (to V.H.), postdoctoral grant from the Umeå Centre for Microbial Research, UCMR (to H.T.), the European Union from the European Regional Development Fund through the Centre of Excellence in Molecular Cell Engineering (2014-2020.4.01.15-0013 to V.H.) and the Estonian Research Council (PRG335 to V.H.). D.N.W. and V.H. groups are also supported by the Deutsche Zentrum für Luft- und Raumfahrt (DLR01Kl1820 to D.N.W.) and the Swedish Research Council (2018-00956 to V.H.) within the RIBOTARGET consortium under the framework of JPIAMR.

## Author contributions
C.C.-M. processed the microscopy data, generated and refined the molecular models and made the structure figures. V.M., K.J.T. and M.K. cloned the ARE constructs, performed genetic manipulations of *E. faecalis* and *L. monocytogenes*, performed polysome fractionations and immunoblotting as well as performed MICs and immunoprecipitations. K.J.T. and V.M. prepared cryo-EM grids. K.V. and J.J. assisted genetic manipulations of *L. monocytogenes*. V.M. collected cryo-EM datasets. H.T. performed genetic manipulations of *B. subtilis* as well as MICs for LsaA variants in *B. subtilis*. M.M. and A.J.O. performed genetic manipulation of *S. aureus* and the generation/characterization of VgaA~LC~ variants. G.C.A. performed sequence conservation analyses. C.P. and Z.I. performed microarray experiments. C.C.-M. and D.N.W. wrote the manuscript with input from all authors. D.N.W. and V.H. conceived and supervised the project.

## Funding

## Competing interests
The authors declare no competing interests.
