## [Peer Review File · Nature Communications]

REVIEWERS' COMMENTS

Reviewer #1 (Remarks to the Author):

Caillan Crowe-McAuliffe and colleagues have addressed most concerns raised by me and the other referee in the rebuttal letter and the revised manuscript. In my opinion there remains one essential point, which the authors have to address. Based on the revised text, which is now much clearer, it has become very apparent that it is invalid to speak of an allosteric mechanism, when referring to the action of ABCF protein on conferring antibiotic resistance. Allostery would imply the binding of a small molecule at a distant site influencing the active site of the protein or molecule it is itself bound to. In this manuscript, a more appropriate way would be to speak of a conformational selection in the PTC, elicited by ABCF binding to the ribosome, which leads to antibiotic egress. However, at the present stage of analysis and the lack of any data, which addresses whether different nucleotide states bound to the ABCF would elicit a similar response on the ribosome, even the claims of conformational selection and especially allostery remain premature. I still maintain that the reconstitution of lysates with hydrolytic mimics of ATP and biochemical analysis of the ability of ABCF to bind the ribosome is the very least to be expected.

Reviewer #2 (Remarks to the Author):

The initial manuscript was largely improved and all our concerns have been addressed or clarified by the authors' replies.

Concerning the authors' response about the hypothetical nature of the state C in the proposed model presented in figure 6 (figure 7 in the revised manuscript). We agree that this state is observed in their structure's reconstruction. Nevertheless, there is no experimental evidence that this state is a relevant post antibiotic release state. It can possibly be a product of the EQ2 mutation.

We congratulate the authors in their effort to better integrate the mutation screen with their structural observations. We think that improves the readability of the manuscript and strengthens their conclusions.

Reviewer #3 (Remarks to the Author):

The manuscript by Crowe-McAuliffe and colleagues reports structural and mutational analysis of how ABCF proteins confer antibiotic resistance to pleuromutilin, lincosamide, and streptogramin A. The ABCF ATPases are becoming a widespread problem in the spread of antibiotic resistance, so a clear understanding of how these proteins carry out their function is needed. The authors show, using three sets of cryo-EM structures, that certain ABCF proteins act by remodeling the peptidyl transferase center (PTC) of the ribosome in a stalled state of initiation, thereby disrupting the antibiotic binding sites. This mechanism of action differs from previously proposed models that the ABCF proteins sterically block the antibiotic binding sites. Notably, simple sequence alignments of the ABCF loop that inserts into the PTC region is not sufficient to identify the mechanism of action and required the structural and mutational analysis presented here. Overall, this is a clearly written study that will be of wide interest and provides a foundation for understanding more of these proteins as they emerge in multidrug resistant pathogens.

Overall, the authors addressed all of the major concerns raised in the previous round of review. There are a few minor changes the authors should make before publication, as described below.

1. On p. 9, please define CTE the first time it is used.
2. In the Methods on p. 41, lines 1142-4 on the use of tRNA microarrays, this seems to be the wrong complex mentioned? Should it be the isolated 70S complex?
3. P. 42, line 1154, should be microliters.
4. Titles for Supplementary Figures S2 and S3 should have "preparation".
5. The red arrows in Supplementary Figures S15-S17 are very hard to see (and would be nearly impossible for color blind individuals). These should be made black.

A separate technical point of interest, but that does not affect the conclusions of the authors is the surprising fact that the multibody refinement did not work better in these cases for the 30S subunit (as shown in Supplementary Figs. S4-S6). Although it's not consequential for this manuscript, it could be worthwhile checking separate focused refinements to see if they improve these regions of the map.

Reviewer #1 (Remarks to the Author)

Caillan Crowe-McAuliffe and colleagues have addressed most concerns raised by me and the other referee in the rebuttal letter and the revised manuscript. In my opinion there remains one essential point, which the authors have to address. Based on the revised text, which is now much clearer, it has become very apparent that it is invalid to speak of an allosteric mechanism, when referring to the action of ABCF protein on conferring antibiotic resistance. Allostery would imply the binding of a small molecule at a distant site influencing the active site of the protein or molecule it is itself bound to. In this manuscript, a more appropriate way would be to speak of a conformational selection in the PTC, elicited by ABCF binding to the ribosome, which leads to antibiotic egress. However, at the present stage of analysis and the lack of any data, which addresses whether different nucleotide states bound to the ABCF would elicit a similar response on the ribosome, even the claims of conformational selection and especially allostery remain premature. I still maintain that the reconstitution of lysates with hydrolytic mimics of ATP and biochemical analysis of the ability of ABCF to bind the ribosome is the very least to be expected.

We have removed the term 'allostery' from the manuscript. As suggested by the reviewer, we now use more explicit language, referring instead to "a conformational selection in the PTC, elicited by ARE binding to the ribosome, leads to antibiotic egress"

Reviewer #2 (Remarks to the Author)

The initial manuscript was largely improved and all our concerns have been addressed or clarified by the authors' replies.

Concerning the authors' response about the hypothetical nature of the state C in the proposed model presented in figure 6 (figure 7 in the revised manuscript). We agree that this state is observed in their structure's reconstruction. Nevertheless, there is no experimental evidence that this state is a relevant post antibiotic release state. It can possibly be a product of the EQ2 mutation.

We agree with the reviewer that the state is possibly affected by the EQ₂ mutation and have mentioned this in the discussion.

We congratulate the authors in their effort to better integrate the mutation screen with their structural observations. We think that improves the readability of the manuscript and strengthens their conclusions.

We thank the reviewers for their careful and thorough examination of our manuscript which led to these improvements in readability and strengthening of the conclusions.

Reviewer #3 (Remarks to the Author)

The manuscript by Crowe-McAuliffe and colleagues reports structural and mutational analysis of how ABCF proteins confer antibiotic resistance to pleuromutilin, lincosamide, and streptogramin A. The ABCF ATPases are becoming a widespread problem in the spread of antibiotic resistance, so a clear understanding of how these proteins carry out their function is needed. The authors show, using three sets of cryo-EM structures, that certain ABCF proteins act by remodeling the peptidyl transferase center (PTC) of the ribosome in a stalled state of initiation, thereby disrupting the

antibiotic binding sites. This mechanism of action differs from previously proposed models that the ABCF proteins sterically block the antibiotic binding sites. Notably, simple sequence alignments of the ABCF loop that inserts into the PTC region is not sufficient to identify the mechanism of action and required the structural and mutational analysis presented here. Overall, this is a clearly written study that will be of wide interest and provides a foundation for understanding more of these proteins as they emerge in multidrug resistant pathogens.

Overall, the authors addressed all of the major concerns raised in the previous round of review. There are a few minor changes the authors should make before publication, as described below.

1. On p. 9, please define CTE the first time it is used.

We have now defined CTE.

2. In the Methods on p. 41, lines 1142-4 on the use of tRNA microarrays, this seems to be the wrong complex mentioned? Should it be the isolated 70S complex?

3. P. 42, line 1154, should be microliters.

4. Titles for Supplementary Figures S2 and S3 should have “preparation”.

We apologize for these thoughtless errors and thank the reviewer for their careful reading of the manuscript. We have now corrected these mistakes.

5. The red arrows in Supplementary Figures S15-S17 are very hard to see (and would be nearly impossible for color blind individuals). These should be made black.

We made the arrows black as requested

A separate technical point of interest, but that does not affect the conclusions of the authors is the surprising fact that the multibody refinement did not work better in these cases for the 30S subunit (as shown in Supplementary Figs. S4-S6). Although it's not consequential for this manuscript, it could be worthwhile checking separate focused refinements to see if they improve these regions of the map.

The multibody refinements did in fact improve the resolution of the small subunits, although unfortunately density for the AREs could not be improved. As a result, maps from the multibody refinements were useful when modelling some regions of the ribosome but not when interpreting our major regions of interest. As a result, we described only the whole volumes in the supplementary materials. The manuscript has been amended to clarify this, and individual components from the multibody refinements are included in the maps deposited in the EMDB.